# Machine Learning for the Multi-Dimensional Bin Packing Problem: Literature Review and Empirical Evaluation

## Abstract

The Bin Packing Problem (BPP) is a well-established combinatorial optimization (CO) problem. Since it has many applications in our daily life, e.g. logistics and resource allocation, people are seeking efficient bin packing algorithms. On the other hand, researchers have been making constant advances in machine learning (ML), which is famous for its efficiency. In this article, we first formulate BPP, introducing its variants and practical constraints. Then, a comprehensive survey on ML for multi-dimensional BPP is provided. We further collect some public benchmarks of 3D BPP, and evaluate some online methods on the Cutting Stock Dataset. Finally, we share our perspective on challenges and future directions in BPP. To the best of our knowledge, this is the first systematic review of ML-related methods for BPP.

## 1 Introduction

With the boom in e-commerce in recent years, logistics is playing a more and more important role in our daily life. An indispensable part of logistics is packing packages into a container or pallet. Traditionally, this task is done by workers, which results in sub-optimal packing configurations. To obtain more compact layouts as well as reduce the cost of packing, the industry is pursuing efficient bin packing algorithms to replace workers with intelligent robotic arms.

As a typical case in combinatorial optimization (CO), the Bin Packing Problem (BPP) has been studied for a long time. Early research can date back to the 1970s (Johnson, 1973; 1974; Coffman et al., 1978a). Like other problems in CO, BPP is well-known to be NP-hard (Garey & Johnson, 1979), indicating that no polynomial time algorithm is currently known for BPP. Due to the tremendous search space in BPP, especially for the 3D version, which is encountered most in reality, we can never apply the exact method (Martello & Vigo, 1998; Martello et al., 2000) to solve this problem within an acceptable time.

In addition to logistics, BPP has other practical applications. Coffman et al. (1978b) extend BPP into multiprocessor scheduling, where the first-fit-increasing algorithm is proposed to find a near-optimal scheduling configuration quickly. Song et al. (2013) model the resource allocating problem in cloud computing as BPP. Concretely, each server is viewed as a bin and each virtual machine as an item. Then, a relaxed online bin packing algorithm is proposed to achieve good performance compared with existing work. Besides, Eliiyi & ELIIYI (2009) mention more applications in the supply chain, including the stock cutting or trim loss problem, project scheduling, financial budgeting, etc. The idea of BPP can be integrated into all these areas with certain adaptations.

On the other hand, the past decade witnessed an explosion in machine learning (ML), especially for deep learning (DL). Since AlexNet (Krizhevsky et al., 2012) won first place by a wide margin in ILSVRC 2012 (Deng et al., 2009), various convolutional neural networks (CNNs) have been proposed to achieve better performance. Other ML techniques like graph neural networks (GNNs) and reinforcement learning (RL) have also made impressive progress in recent years. Nowadays, these techniques can be adopted in a large number of fields, e.g. medicine, computer vision, and autonomous driving.

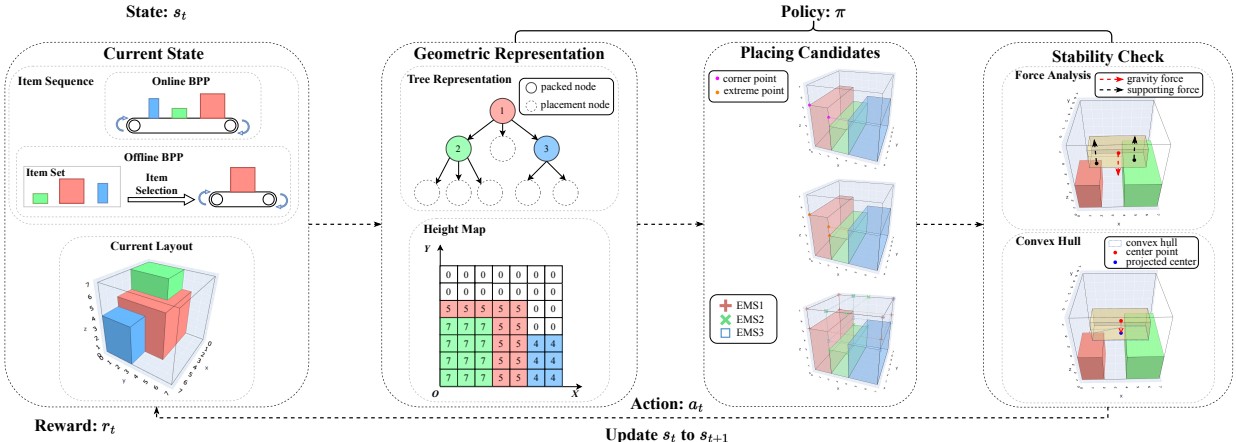

Figure 1: A typical pipeline of ML-related methods for 3D BPP. We derive the geometric representation of the current state, determine placing candidates, and check the stability before placing an item. ML-related methods usually encode rich geometric representations of the current state while traditional ones tend to evaluate it by simple man-made rules which cannot precisely estimate its value and lack generalizability. In modern scenes, the action of placing an item from the conveyor belt into the bin is conducted by a robotic arm rather than a worker.

Owing to the rapid development in ML and the imperative need for efficient bin packing algorithms, it is natural for us to try applying ML techniques to BPP. This approach has several advantages over traditional search-based methods:

**Efficiency.** ML techniques often involve matrix multiplication and convolution, which can be computed in parallel on GPUs efficiently, while traditional methods rely on CPUs to execute iterative and serial operations, which can hardly exploit the computational power of GPUs. Moreover, the runtime of many traditional methods rises exponentially with the growth of the problem scale, but the additional overhead of ML techniques is much smaller due to parallel computation on GPUs.

**Data-driven characteristic.** With the development of computing power and simulation platforms, one can easily generate extensive data from different distributions, and train ML models in a simulation environment similar to reality. During training, these models can explore the patterns in the data, and also interact with the simulation environment to achieve better performance.

**Less reliance on domain knowledge.** Traditional methods require experts to determine an explicit strategy, e.g. score function, to compare one solution with another, whereas learning-based methods do that in a much more implicit way by its internal mechanism, e.g. neural network, which is trained by data.

Actually, there are already a number of studies on combining ML with CO problems, e.g. Travelling Salesman Problem (TSP). Vinyals et al. (2015) propose a new architecture named Pointer Net, which leverages the attention mechanism as a pointer to the input sequence. The pointer will select a member of the input sequence as an output at each step and finally produce a permutation of the input sequence. More examples of the incorporation of ML and CO can be found in recent surveys (Bengio et al., 2021; Yan et al., 2020; Mazyavkina et al., 2021). However, these surveys cast limited sight on learning for BPP, to say nothing of a systematic review on that. Hence, unlike previous surveys on BPP, which investigate the traditional or heuristic methods (Christensen et al., 2017; Coffman et al., 2013), this survey will shed light on ML-related ones and put forward some future directions. We can briefly divide existing bin packing algorithms into three classes: traditional, learning-based, and hybrid.

## 2 Preliminaries

In this section, we introduce the Bin Packing Problem (BPP) and its variants. In the meantime, we give a brief overview of traditional methods and practical constraints in this problem.

$$
\min \ \gamma
$$

$$
\text{s.t.}
\begin{cases}
\delta_{i1} + \delta_{i2} + \delta_{i3} + \delta_{i4} + \delta_{i5} + \delta_{i6} = 1 & (1a) \\
l_{ij} + l_{ji} + u_{ij} + u_{ji} + b_{ij} + b_{ji} + c_{ij} + c_{ji} = 1 & (1b) \\
x_i - x_j + L \cdot (l_{ij} - c_{ij} - c_{ji}) \leq L - l'_i & (1c) \\
y_i - y_j + W \cdot (u_{ij} - c_{ij} - c_{ji}) \leq W - w'_i & (1d) \\
z_i - z_j + H \cdot (b_{ij} - c_{ij} - c_{ji}) \leq H - h'_i & (1e) \\
(l_{ij} + l_{ji} + u_{ij} + u_{ji} + b_{ij} + b_{ji})(\Gamma - 1) + \gamma_i - \gamma_j + c_{ij}\Gamma \leq \Gamma - 1 & (1f) \\
0 \leq x_i \leq L - l'_i & (1g) \\
0 \leq y_i \leq W - w'_i & (1h) \\
0 \leq z_i \leq H - h'_i & (1i) \\
0 < \gamma_i \leq \gamma \leq \Gamma & (1j) \\
l'_i = \delta_{i1}l_i + \delta_{i2}l_i + \delta_{i3}w_i + \delta_{i4}w_i + \delta_{i5}h_i + \delta_{i6}h_i & (1k) \\
w'_i = \delta_{i1}w_i + \delta_{i2}h_i + \delta_{i3}l_i + \delta_{i4}h_i + \delta_{i5}l_i + \delta_{i6}w_i & (1l) \\
h'_i = \delta_{i1}h_i + \delta_{i2}w_i + \delta_{i3}h_i + \delta_{i4}l_i + \delta_{i5}w_i + \delta_{i6}l_i & (1m) \\
\delta_{i1}, \delta_{i2}, \delta_{i3}, \delta_{i4}, \delta_{i5}, \delta_{i6} \in \{0, 1\} & (1n) \\
\gamma_i, \gamma \in \mathbb{Z} & (1o) \\
l_{ij}, u_{ij}, b_{ij}, c_{ij} \in \{0, 1\} & (1p)
\end{cases}
\tag{1}
$$

### 2.1 Problem Formulation

In general, BPP can be categorized into 1D, 2D, and 3D BPP by the dimension of the items to be packed. In the logistics industry, 3D BPP is the most common problem. Consequently, we will introduce 3D BPP in detail. The other two forms can be reasoned by analogy.

Typically, a set of cuboid items with their length ($l_i$), width ($w_i$), and height ($h_i$) is given. Then, we need to pack all these items into one or multiple homogeneous cuboid bins. Each bin's measurements are ($L, W, H$), representing its length, width, and height respectively. It's worth noting that the objective varies according to practical needs. For convenience, we suppose that the objective is to minimize the number of used bins. We denote ($x_i, y_i, z_i$) as the left-bottom-back corner of item $i$ and $(0, 0, 0)$ as the left-bottom-back corner of the bin. Then, BPP can be formulated as Equation 1.

Constraints $1a, 1g$ to $1o$ are subject to $i = 1, \ldots, n$ and others subject to $i, j = 1, \ldots, n, i \neq j$. $\delta_{i\cdot}$ means the orientation of item $i$ because there are 6 orientations leading to different permutations of measurements. $\gamma_i$ means which bin item $i$ is inside and $\Gamma$ means the number of usable bins. ($l_{ij}, u_{ij}, b_{ij}$) indicates whether item $i$ is to the left, under and back of item $j$. $c_{ij}$ indicates whether the bin number of item $i$ is less than that of item $j$. ($l'_i, w'_i, h'_i$) indicates the measurements of item $i$ after rotation defined by $\delta_{i\cdot}$.

Constraints $1c$ to $1e$ ensure that any two items will not overlap and Constraints $1g$ to $1j$ guarantee that every item is inside exactly one bin.

A small example is shown in Figure 2. We can see that none of the colored items reaches outside the bin and that any two items do not overlap. Hence, it is a valid solution to this problem instance.

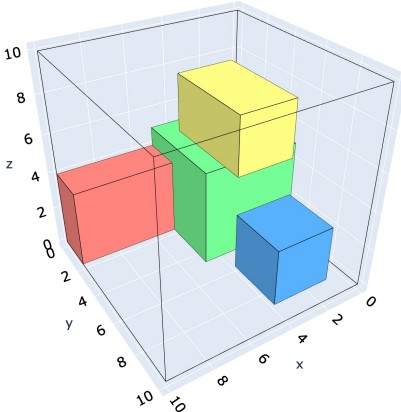

Figure 2: A simple example of 3D BPP. Four items with size $(5, 5, 5)$ (green), $(4, 2, 5)$ (red), $(3, 3, 3)$ (blue) and $(5, 3, 3)$ (yellow) are given. After certain rotations, they are placed in a $(10, 10, 10)$ cuboid bin sketched by black lines. The green item with measurements $(5, 5, 5)$ (after rotated) is located at $(0, 0, 0)$. The red item with measurements $(5, 2, 4)$ is located at $(5, 0, 0)$. The blue item with measurements $(3, 3, 3)$ is located at $(1, 6, 0)$. The yellow item with measurements $(3, 5, 3)$ is located at $(1, 1, 5)$.

## 2.2 Problem Variants

Generally, the Bin Packing Problem is divided into two classes: online and offline, according to our prior knowledge about items to come (refer to Figure 1 for illustration). Besides, there are a variety of practical constraints we need to take into account when applying bin packing algorithms to real-world settings.

**Online BPP.** It lays the restriction that one can only know the information of the current item to be packed, and must pack items in an unknown sequence. This setting is common in assembly lines, where workers make an instant decision when accessing the item. Without the knowledge of subsequent items, we can hardly obtain the global optimal solution. Instead, we tend to seek a local optimal solution, following some heuristic strategy.

Researchers put emphasis on the easiest 1D BPP at the early stage. Next-fit, first-fit and best-fit (Johnson, 1973) are among the most popular methods for 1D online BPP. The ideas are quite simple. Next-fit tries to pack the current item into the current bin. If the bin cannot contain the item, it will close the current bin and open a new bin for the item. First-fit can be viewed as an improved version of next-fit because it will not close the previous bins. Instead, the current item will be packed into the lowest indexed bin that can contain the item. Only when none of the non-empty bins can contain the item will a new bin be opened. Best-fit revises first-fit by packing the current item into the bin with the least remaining space.

The heuristics above can help determine which bin to select but the placing location remains unsolved in BPP with higher dimension. For 2D BPP, Jakobs (1996) design the bottom-left (BL) heuristic that shifts the current item from the top as far as possible to the bottom and then as far as possible to the left. Hopper & Turton (1999) point out that Jakobs's algorithm may cause big vacancy in the layout. Thus, they propose the bottom-left-fill (BLF) heuristic to mitigate the problem. Berkey & Wang (1987) build levels splitting the rectangular bin. Therefore, each level is regarded as a 1D bin, and 1D heuristics can be used. Similarly, these heuristics can be adapted to 3D BPP. For instance, Deepest Bottom Left with Fill (DBLF) (Karabulut & İnceoğlu, 2004) and Bottom-Left-Back-Fill (BLBF) (Tiwari et al., 2008) are tailored versions for 3D BPP by considering shift along $Z$ axis.

To evaluate a packing algorithm $\mathcal{A}$, researchers have introduced the asymptotic performance ratio (Seiden, 2002) as:

$$R_{\mathcal{A}}^{\infty} = \limsup_{n \to \infty} \sup_{\sigma} \left\{ \frac{\text{cost}_{\mathcal{A}}(\sigma)}{\text{cost}(\sigma)} \mid \text{cost}(\sigma) = n \right\}, \tag{2}$$

where $\sigma$ denotes the item sequence and $\text{cost}(\sigma)$ the minimum possible number of used bins. It proves that the ratios for next-fit, first-fit, and best-fit are 2, 1.7, and 1.7, respectively (Johnson et al., 1974; Coffman et al., 1984), which implies the obvious sub-optimality of traditional methods.

**Placing Candidates.** The aforementioned methods like BLF merely consider one placing candidate. To obtain a more compact layout, it needs to involve more placing candidates. Since the space is continuous, we cannot enumerate all points. To reduce the search space and maintain the performance, heuristics for computing placing candidates are designed. Martello et al. (2000) select corner points from the layout. All points that no placed item has some part right of or above or in front of can form an outer space. Then, the border splitting the outer space and items is defined as "envelope". Points where the slope of "envelope" changes from vertical to horizontal are the so-called corner points. Crainic et al. (2008) propose the concept of extreme points. When an item with size $(l_i', w_i', h_i')$ after rotated is placed at $(x_i, y_i, z_i)$, point $(x_i + l_i', y_i, z_i)$ will be projected along $Y$ and $Z$ axis onto the nearest item or wall. Similar projection of point $(x_i, y_i + w_i', z_i)$ and point $(x_i, y_i, z_i + h_i')$ can be inferred with ease. The projection points are the new extreme points. Empty maximal-space (EMS) is introduced to BPP in (Gonçalves & Resende, 2013). Here EMS denotes the largest empty orthogonal spaces available for filling with items. Then, corners of EMS can be viewed as placing candidates. Refer to Figure 1 for comparison.

**Offline BPP.** It provides us with the information of all items to be packed and the full manipulation of item sequence. Since we may determine the sequence items come in, we can obtain a globally optimal solution theoretically. Chen et al. (1995) formulate the 3D BPP as a mixed-integer programming (MIP) model. Subsequently, it can be solved by commercial solvers like Gurobi (Gurobi Optimization, LLC, 2023) and CPLEX (Cplex, 2009). Martello et al. (2000) have described an exact branch-and-bound algorithm for 3D BPP. They first assign items to different bins through a main branching tree. Then, the branch-and-bound algorithm is used to determine the placing locations. Pruning happens when items belonging to a bin cannot be packed in it or the total volume will not improve even if we make full use of the remaining space.

The drawback of the MIP model and the exact algorithm is their high time complexity. When the scale of problems is large, they cannot generate satisfactory solutions within a reasonable time (Wu et al., 2010; Laterre et al., 2019). Hence, researchers have been exploring another way out. BPP can be viewed as a grouping problem (Falkenauer et al., 1992). They augment the standard genetic operator with a group part so that crossover, mutation, and inversion operators fit this grouping problem. Inspired by the Dominance Criterion (Martello & Toth, 1990), Falkenauer (1996) further improve the above Grouping Genetic Algorithm (GGA) and propose hybrid GGA.

Many other traditional search-based methods are also incorporated. Guided Local Search (GLS) is used in (Faroe et al., 2003). At each step, GLS guides the current layout to its neighbor, where one item is reallocated in the original bin or another. The iteration stops when the local minimum of the objective function is found. The work (Li et al., 2018) based on Monte-Carlo Tree Search (MCTS) utilizes the selection, expansion, simulation, and backpropagation to explore states leading to high volume utilization. Besides, a Top-K table is designed to prune the tree. Starting from an initial solution, Lodi et al. (1999) try to remove a bin and transfer items inside to other bins with tabu search (Glover & Laguna, 1998). Though these methods accelerate the searching process to some extent, they may still fail to meet realistic needs in a limited time.

**Misc.** There are other variants of BPP, which can be seen as derivatives of online BPP or offline BPP. For example, some scenes require that the items are first picked out from a bin, and then packed into another (Hu et al., 2020; Tanaka et al., 2020). Verma et al. (2020) enhance online BPP by providing the information of several upcoming items.

### 2.3 Practical Constraints

In reality, there is a wide range of practical constraints we need to consider when dealing with BPP. For example, when a robotic arm is used to place items, the algorithm should guide the arm to avoid collisions. Bortfeldt & Wäscher (2013) carefully investigate practical constraints existing in a variant of 3D BPP, i.e. Container Loading Problem (CLP). They classify constraints into container-related ones, item-related ones, cargo-related ones, positioning ones, and load-related ones, each of which contains more fine-grained

Table 1: Comparison between different learning-based methods. CNN and attention mechanism are the two most popular modules while stepwise form overwhelms terminal one in reward function in case of sparse reward. Various state representations, stability check methods, and RL skills are applied in these works. Note that space utilization represents a set of metrics concerned with how well the placements make use of the bin space. Abbreviations: actor-critic (AC) (Konda & Tsitsiklis, 1999), generalized advantage estimation (GAE) (Schulman et al., 2015), proximal policy optimization (PPO) (Chung et al., 2014), actor critic using Kronecker-factored trust region (ACKTR) (Wu et al., 2017), prioritized oversampling (PO) (Zhang et al., 2021), and advantage actor-critic (A2C) (Mnih et al., 2016).

| Approach | Problem type | Dimension | Module | State | Reward | Optimization objective | Stability check | RL skill |
|---|---|---|---|---|---|---|---|---|
| Li et al. (2020) | offline | 3D | attention | item information | stepwise | space utilization | none | AC + GAE |
| Zhao et al. (2021) | online | 3D | CNN | height map + item size | stepwise | space utilization | supporting area + corner support | AC |
| Tanaka et al. (2020) | misc. | 3D | FCN | height map | stepwise | space utilization | supporting area | PPO |
| Hu et al. (2020) | misc. | 2D + 3D | CNN + GRU + attention | precedence graph + height map + item size | terminal | space utilization + stable ratio | convex hull | AC |
| Zhao et al. (2022) | online | 3D | CNN | height map + item size + feasibility mask | stepwise | space utilization | convex hull + force analysis | ACKTR |
| Zhang et al. (2021) | offline | 3D | CNN + attention | item size + frontier | terminal | space utilization | none | PO |
| Jiang et al. (2021b) | offline | 3D | CNN + attention | item information + height map | stepwise | space utilization | none | A2C + GAE |

restrictions, e.g. stability, weight distribution, orientation, and weight limit. Among all these constraints, stability may be the most significant in bin packing applications. Stability ensures every item can remain static against gravity throughout the whole packing process. Otherwise, toppling down may occur, which can damage items or hurt people, causing huge economic losses and safety hazards in the industry. Nevertheless, the complexity of the stability problem increases in proportion to the number of items because of the hardness of modeling.

Efforts made to tackle the stability problem can be broadly classified into two types: i) intuitive methods (Gzara et al., 2020; Laterre et al., 2019; Zhao et al., 2021) either use the supporting area or firm support at particular points, e.g. the center point or corners, to check the stability. A wiser criterion may be to judge whether the center of mass is within the convex hull of supporting points (Ramos et al., 2016a); ii) theoretical methods (Ramos et al., 2016b) check stability by verifying whether each item satisfies the force and moment balance condition derived from Newton's laws while Wang & Hauser (2019) also take friction into consideration. Comparison is shown in Figure 1. Though the stability problem is of great importance to practical BPP, most works either neglect this problem or model it in a very simple way, which hinders their practical usage.

## 3 Machine Learning for the Bin Packing Problem

Although heuristics mentioned in Section 2 speed up the searching process, they still cannot find a good enough solution within a reasonable time when the problem scale increases because those methods tend to fall into a local optimum. Another concern is that expert knowledge, which counts for traditional methods, is sometimes hard to obtain. Therefore, ML has been introduced into BPP. In this section, we will make a literature review on learning-based and hybrid methods. The former represents methods that only leverage ML techniques to accomplish all tasks in BPP while the latter refers to methods infusing ML skills with conventional heuristics. To the best of our knowledge, only a little existing research has been conducted on ML for BPP. We list almost all the ML-related methods in Table 1 and 2. Obviously, 3D BPP receives the most attention due to its practical value. All listed methods exploit an RL framework while reserving differences in model configuration. The high-level view of the RL framework is provided in Figure 1. The current state $s_t$ consists of the current layout and current item. In each iteration, the agent picks the best action $a_t$ according to its policy $\pi$. Actions failing to pass the stability check are excluded from $\pi$. Then, the state $s_t$ is updated to $s_{t+1}$, and the reward $r_t$ is calculated. We assume the readers are knowledgeable about the basic concepts in recent deep nets, while we refer to (Goodfellow et al., 2016) for some preliminary concepts.

### 3.1 Learning-Based Methods

Learning-based methods have been applied to different forms of BPP, i.e. online, offline, and miscellaneous versions.

**Models for Online BPP.** Zhao et al. (2021) discretize the 2D space and introduce an $L \times W$ height map to record the maximum height on each grid. The size of the current item is expanded into an $L \times W \times 3$ tensor and encoded together with the height map by a CNN. Then, a CNN predicts the probability of each action whilst another CNN eliminates illegal actions. Later, they extend their work in three aspects (Zhao et al., 2022): i) utilize a stacking tree to check stability according to static equilibrium analysis, raising both accuracy and efficiency; ii) decompose the actor network into three serial components to reduce the action space; iii) design a specialized reward function to encourage the robotic arm to place items from far to near, aiming for collision-free trajectories.

**Models for Offline BPP.** Generally, offline BPP can be decomposed into three sub-actions: item selection, orientation selection, and position selection, done either successively or concurrently. The problem can be regarded as a Markov Decision Process (MDP) if they are done concurrently. Otherwise, the last two steps are not strict MDP because they are conditioned on previous sub-actions. To address this problem, Li et al. (2020) utilize the prediction of the last sub-action as a conditional query for the current attention-based decoder to formulate a strict MDP. However, a potential weakness in this work is its neglect of the geometric representation of placed items.

Frontier, a front-view variant of the height map, is adopted as the geometric representation (Zhang et al., 2021). They reduce the action space by only inferring orientation and $Y$ coordinate with state embeddings and also propose prioritized oversampling to learn on hard examples again. Experiments show that they achieve state-of-the-art performance in offline 3D BPP.

**Models for Miscellaneous Scenes.** A more complicated scene is studied in (Tanaka et al., 2020): items must be transferred from a bin to a tote. They first parameterize both bin and tote as height maps with shape $L_B \times W_B$ and $L_T \times W_T$. Then, two maps are expanded and concatenated into an $L_B \times W_B \times L_T \times W_T \times 2$ tensor as the input of a fully convolutional network (FCN) (Long et al., 2015). Finally, the network will output the probability of each action. Though they demonstrate that simultaneous planning for item picking and placing outperforms individual one, the action space becomes prohibitively large when the sizes of the bin and tote increase.

Hu et al. (2020) focus on a similar transportation scene. Different from (Tanaka et al., 2020), they utilize a precedence graph to describe the picking dependency among items, which makes clear how to pick up a particular item. At each step, the encoder encodes size information and the precedence graph of several items. Then, with the height map and attention module, the decoder infers the next item to pack and its orientation. The packing location is determined by a GRU.

### 3.2 Hybrid Methods

Machine learning skills are so versatile and flexible that they can easily coordinate with conventional heuristics. Existing works on BPP mainly focus on the following three directions:

**Combining RL with the Packing Heuristic (PH).** Inspired by (Bello et al., 2016; Vinyals et al., 2015), Hu et al. (2017) are the first to introduce the pointer mechanism along with RL to BPP. Specifically, they select items with pointer and leave orientation and position selection to an EMS-based heuristic. Later, Duan et al. (2018) also include orientation selection in the learning architecture by adding an intra-attention module. Moreover, they employ a multi-task framework to mitigate the imbalance among the three training procedures.

A critical point in ML-related methods is how to represent the bin state. It is hard for learning models to understand the geometric representation with mere positions and sizes of placed items, so researchers make efforts to enrich this representation. Despite the popularity and concision of the height map as the geometric representation, it ignores the mutual spatial relationship among placed items and receives criticism against discrete solution space. (Zhao & Xu, 2022) tackles this by viewing each placed item and each placing

Table 2: Comparison between different hybrid methods. Different from Table 1, hybrid methods mainly focus on offline BPP, in which additional variable, i.e. item sequence, needs to be resolved. Other meaningful objectives, e.g. surface area and bin quantity, are considered in this domain. One notable defect is lack of stability check, which restricts their practicality. Abbreviation: policy gradient (PG) (Sutton et al., 1999), beam search (BS), deep Q-network (DQN) (Mnih et al., 2013), constraint programming (CP), prioritized experience replay (PER) (Schaul et al., 2015), and neural network (NN).

| Approach | Problem type | Dimension | Module | State | Reward | Optimization objective | Stability check | RL skill |
|---|---|---|---|---|---|---|---|---|
| Hu et al. (2017) | offline | 3D | pointer mechanism + packing heuristic (PH) | item size | terminal | surface area | none | PG + BS |
| Duan et al. (2018) | offline | 3D | pointer mechanism + intra-attention + PH | item size | terminal | surface area | none | PPO |
| Verma et al. (2020) | misc. | 3D | DNN + PH | height map + border encoding + placement vector | stepwise | bin quantity | none | DQN |
| Cai et al. (2019) | offline | 1D | simulated annealing | item assignment | stepwise | bin quantity | none | PPO |
| Laterre et al. (2019) | offline | 2D + 3D | DNN + MCTS | item information | terminal | surface area | center support | self-play |
| Goyal & Deng (2020) | offline | 3D | CNN + PH | voxel representation | stepwise | space utilization | none | PPO |
| Jiang et al. (2021a) | offline | 3D | CNN + attention + CP | item information + height map | stepwise | space utilization | none | A2C + GAE |
| Zhao & Xu (2022) | online | 3D | pointer mechanism + GATs + PH | item information + placement vector | stepwise | space utilization | convex hull + force analysis | ACKTR |
| Puche & Lee (2022) | online | 3D | CNN + MCTS | height map + item size | stepwise | space utilization | supporting area + corner support | PER |
| Yang et al. (2023) | online | 3D | CNN | item size + voxel grid | stepwise | space utilization | supporting area + NN prediction | PPO |

candidate as a tree node (shown in Figure 1). Hence, relative position relations are embedded in edges. They further encode the tree with graph attention networks (GATs) (Veličković et al., 2017) and select a heuristic placing candidate in continuous solution space with the pointer mechanism. Results show that this method achieves state-of-the-art performance in online 3D BPP. Yang et al. (2023) represent the current layout as the 3D voxel grid to capture the wasted space. A neural network trained with recorded data is used to predict the stability of layouts. Besides, they propose an unpacking heuristic to unpack placed items to further improve space utilization.

Instead of the pointer mechanism, Goyal & Deng (2020) exploit a CNN and its Softmax operator to select items, followed by the BLBF heuristic to determine orientation and position as aforementioned. Another way is to propose placements with a packing heuristic and choose the best from them with respect to each placement's value estimated by RL (Verma et al., 2020). The advantage of combining RL with the packing heuristic is reducing the search space of RL because RL is only responsible for partial tasks in BPP.

**Combining RL with the Heuristic Search.** Heuristic search can provide the RL agent with additional reward signals, thus speeding up the training process. The exploration nature of heuristic search also helps to improve performance. Cai et al. (2019) initialize a complete assignment randomly. Then the RL agent swaps items to generate an initial solution, followed by further optimization steps conducted by simulated annealing. MCTS is used to execute expansion and simulation on a selected initial packing configuration (Laterre et al., 2019). The state transitions are stored along with a ranked reward computed by comparing current and recent performance. Subsequently, the network is updated with these records. Such a self-play manner can encourage the agent to keep surpassing itself. Puche & Lee (2022) harness MCTS with rollouts to form a model-based method. They augment the data with rotation and flip transformations, and train the agent with prioritized experience replay (Schaul et al., 2015). Jiang et al. (2021a) integrate RL with a constraint programming (CP) solver for better solutions. They set orientation and position as decision variables and apply a branch-and-bound algorithm to them, during which the RL agent will decide on their values at each step in CP.

**Combining SL with the Heuristic.** Besides RL, Supervised Learning (SL) can also be fused with BPP. Mao et al. (2017) extract features of items and bins manually. According to these features, the most suitable heuristic to pack items is selected by a neural network, which is trained with real-world logistics orders. Chu & Lin (2019) pretrain a CNN with labeled data generated by heuristic methods and fine-tune it through RL. Duan et al. (2018) treat the current best solution as ground truth to train the orientation selection module with the help of a hill-climbing algorithm. To reduce the action space, Jiang et al. (2021a) infer

Table 3: The bin size and ranges of item sizes in (Martello et al., 2000).

| Class | Bin size $(L, W, H)$ | Item Length | Item Width | Item Height |
|---|---|---|---|---|
| 1 | (100,100,100) | $[1, L/2]$ | $[2W/3, W]$ | $[2H/3, H]$ |
| 2 | (100,100,100) | $[2L/3, L]$ | $[2W/3, W]$ | $[1, H/2]$ |
| 3 | (100,100,100) | $[2L/3, L]$ | $[1, W/2]$ | $[2H/3, H]$ |
| 4 | (100,100,100) | $[L/2, L]$ | $[W/2, W]$ | $[H/2, H]$ |
| 5 | (100,100,100) | $[1, L/2]$ | $[1, W/2]$ | $[1, H/2]$ |
| 6 | (10,10,10) | $[1, 10]$ | $[1, 10]$ | $[1, 10]$ |
| 7 | (40,40,40) | $[1, 35]$ | $[1, 35]$ | $[1, 35]$ |
| 8 | (100,100,100) | $[1, 100]$ | $[1, 100]$ | $[1, 100]$ |

an action embedding instead. Then, an approximate function trained via SL will map the embedding to a position. Zhu et al. (2021) train a pruning network with historical branching choices to guide the tree search for potentially valuable states. Although SL accelerates the training, the accessibility or quality of training data usually remains a problem, hindering its applicability.

## 4 Benchmarks

When talking about benchmarks in BPP, one may refer to the BPP library BPPLIB (Delorme et al., 2018). However, it only contains 1D BPP instances, which are inconsistent with the realistic scenes. Public benchmarks in 3D BPP are relatively scarce and short of cognition. Since researchers are inclined to test their algorithms on their self-made datasets, it is difficult for us to compare different algorithms' performance. To facilitate future research, we introduce three comparatively well-developed benchmark datasets and a virtual environment below.

**Random Dataset (RD).** An early dataset can be traced back to (Martello et al., 2000), which contains altogether 9 classes of random instances. The first 8 classes uniformly sample instances from corresponding size ranges respectively while instances in the last class are generated by cutting three bins into small parts recursively. The distribution of RD[1] is completely set by simple rules, so it may not reflect the real-world situation. The information of the first 8 classes is listed in Table 3.

**Synthetic Industrial Dataset (SID).** Elhedhli et al. (2019) realize the aforementioned distribution issue, so they analyze some statistics, i.e. item volumes, item repetitions, height-width ratio, and depth-width ratio, of industrial data. Then, they utilize the closest pre-defined distribution to approximate the true one. It is proved that SID[2] is not significantly different from the real-life dataset. Hence, models trained on this artificial dataset may generalize well to realistic scenes.

**Cutting Stock Dataset (CSD).** Akin to the ninth class in RD, CSD[3] is generated by cutting the bin into items of pre-defined 64 types (Zhao et al., 2021). Then, these items are reordered either by their *Z* coordinates or stacking dependency: before an item's enqueueing, all of its supporting items should be added to the sequence. Hence, CSD has a remarkable advantage over RD and SID: the achievable upper bound of occupancy rate (i.e. ratio of placed items' total volume to bin volume) is 100%. Such property gives researchers an intuitive view of how well their algorithms perform.

**PackIt.** PackIt[4] (Goyal & Deng, 2020) is a benchmark and virtual environment for geometric planning. This task can be regarded as a special 3D offline BPP, where items have irregular shapes, collected from ShapeNet (Chang et al., 2015), other than cuboids. Besides, an interactive environment based on Unity (Goldstone, 2009) is built for training the RL agent and physical simulation. For convenience, PackIt

---

[1]`http://hjemmesider.diku.dk/~pisinger/codes.html`
[2]`https://github.com/Wadaboa/3d-bpp`
[3]`https://github.com/alexfrom0815/Online-3D-BPP-DRL`
[4]`https://github.com/princeton-vl/PackIt`

also provides a heuristic-based and a hybrid model, mentioned in Section 3.2, as baselines for researchers to compare their algorithms with. Some details of the environment are listed as follows:

**Tasks:** i) Selecting an item to pack; ii) Selecting its rotation; iii) Selecting packing position for the rotated item;

**Reward:** The ratio of the current placed item's volume to the total volume of all items in a sample.

**Metrics:** i) Average cumulative reward across all samples in the dataset; ii) Percentage of samples in the dataset achieving a cumulative reward not less than a given threshold.

## 5 Experiments

Since most methods tune their methods on different datasets, we only test four online methods on the Cutting Stock Dataset described in Section 4. PCT (Zhao & Xu, 2022) and BPPDRL (Zhao et al., 2021) are ML-related methods, while OnlineBPH (Ha et al., 2017) and 3DBP[5] are heuristic methods with superior performance.

As shown in Table 4, PCT outperforms OnlineBPH and 3DBP in space utilization by a large margin, whether stability is guaranteed or not. Also, the time cost of PCT is close to heuristic methods. Hence, ML-related methods have the potential to be applied in reality. We do not compare ML-related methods with commercial solvers, because the item information is given in sequence and instant decisions should be made in 3D online BPP. Therefore, it cannot be solved by MIP.

Table 4: Performance of different methods. The space utilization rate (Util.) and time cost per item are listed for comparison.

| Method | With stability | | Without stability | |
|---|---|---|---|---|
| | Util. | Time (s) | Util. | Time (s) |
| PCT | **0.753** | $2.32 \times 10^{-2}$ | **0.828** | $6.65 \times 10^{-3}$ |
| BPPDRL | 0.660 | $1.34 \times 10^{-2}$ | / | / |
| OnlineBPH | 0.627 | $1.85 \times 10^{-2}$ | 0.672 | $1.12 \times 10^{-2}$ |
| 3DBP | / | / | 0.689 | $1.54 \times 10^{-3}$ |

## 6 Challenges and Future Directions

Existing research on ML for BPP has manifested its potential. A large part of it achieves competitive or even better performance than traditional methods in a shorter running time. Also, the data-driven characteristic and less reliance on domain knowledge of ML are appealing. Unfortunately, there are still challenges before applying these ML-related methods to practice. In this section, we will discuss underlying challenges and future directions in this promising field.

**Practical Feasibility.** As shown in Table 1 and 2, more than half of the ML-related methods neglect the physical stability problem, which is probably the most significant factor weakening their practical feasibility. In real-life scenes like logistics, 3D BPP should be treated as a Constrained Markov Decision Process (CMDP) (Altman, 1999) rather than MDP. This will bring some trouble because standard ML cannot guarantee to engender a solution satisfying such hard constraints. Zhao et al. (2021) and Yang et al. (2023) leverage a neural network to predict actions leading to stable layouts. Hu et al. (2020) integrate stability into the reward function. These two methods reduce the probability of instability, but they still cannot eliminate it. Some works (Zhao et al., 2022; Zhao & Xu, 2022; Tanaka et al., 2020; Laterre et al., 2019) model the stability constraint and directly mask out actions leading to unstable layouts before sampling. Other constraints can also be satisfied through such a method if modeled accurately, but it may lead to instability in training. Another concern is the reachability of the robotic arm. Due to the degree of freedom

---

[5]https://github.com/jerry800416/3D-bin-packing

and arm length, some placements cannot be realized. Worse still, to avoid collisions, industries prefer to place items from a tilted angle rather than in a top-down direction, which means middle positions lower than surroundings are harder to reach. Sometimes, modeling such hard constraints is non-trivial. A potential solution is to train the RL agent with a physics engine, e.g. Bullet or MuJoCo, so the agent will automatically know whether it violates constraints and does backpropagation on this signal. More insights on handling CMDP problems can be found in (Liu et al., 2021). In reality, the offline algorithm should be able to replan in case items come in a sequence different from the schedule due to random errors in the assembly line. A simple solution is to combine offline algorithms with online ones so that an instant decision will be made by online algorithms when errors occur.

**Demand for Datasets.** Despite the existence of a few public datasets mentioned in Section 4, they lack enough popularity owing to loss of integrity or authenticity. Consequently, an authoritative 3D BPP dataset is in need for further research. (See the significance of ImageNet to the object recognition field.) On the other hand, information about the item size alone is insufficient for practical use. In the logistics industry, extra prior knowledge, e.g. item weight, weight distribution and fragility, can be as vital as size. Motivated by this, we believe collecting an elaborate industrial dataset will pave the way for practical applications of ML-related methods. In addition, we can augment the dataset with a virtual environment or theoretical criteria, e.g. inverse kinematics and force analysis, to inform us whether a certain placement is viable.

**Selection of Learning Architectures.** In recent years, GNN has been applied in several CO fields, e.g. TSP and minimum spanning trees (MST), due to their inherent graph structure. Likewise, we can represent the topological structure of placed items in the bin as a directed graph. Various types of GNN (Wu et al., 2020) can then be used to extract features from the graph. Among methods mentioned in Section 3, only Zhao & Xu (2022) leverage the graph structure in BPP. Therefore, there is still much space to explore.

Imitation learning may be another wise choice when we have access to packing data from experienced workers in the industry. With expert demonstrations, we can approximate the expert's reward function and train our agent through methods like inverse reinforcement learning (Abbeel & Ng, 2004).

## 7 Conclusion

This article first gives a brief overview of BPP and traditional solvers. Then a systematic review of ML-related methods for multi-dimensional BPP is provided, especially on 3D BPP which is an emerging topic in ML research with vital practical importance. The advantages of ML-related methods include efficiency, data-driven characteristic, and less reliance on domain knowledge. Also, public benchmarks in 3D BPP are summarized, and empirical evaluation on 3D online BPP is conducted. Finally, we discuss challenges and future directions in this field to stimulate future study and industrial applications.

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
