# OpenReview forum: "Machine Learning for the Multi-Dimensional Bin Packing Problem: Literature Review and Empirical Evaluation"
_TMLR — Withdrawn by Authors_

### Review · Reviewer_T99e · 2024-01-22

**Summary Of Contributions:**

This paper focuses on the combinatorial optimization problem of 2D and 3D Bin Packing. With the goal of surveying the existing use of machine learning to solve this combinatorial optimization problem, this paper first presents the complete 3D Bin Packing problem and its online and offline variants, discussing various non-machine-learning based combinatorial algorithms for this problem. After briefly presenting practical constraints pertaining to stability, the paper covers machine learning schemes for the different variants of the Bin Packing problem. The authors categorize these methods between purely machine learning based methods, and hybrid methods that leverage both machine learning and some problem specific heuristic. After presenting the methods, the paper covers some benchmarks for multi-dimensional Bin Packing, and evaluates 2 machine learning based methods and 2 heuristic methods on one of the benchmarks, highlighting the advantage that machine learning can provide on this problem of 3D Bin Packing. The authors then discuss the open challenges and future directions for machine learning based multi-dimensional Bin Packing, covering important aspects such as the necessity of considering practical constraints such as stability in the learning, or the need for more datasets and benchmarks to appropriately evaluate existing and future schemes.

**Audience:**

Yes

**Broader Impact Concerns:**

No broader impact concerns

**Claims And Evidence:**

No

**Requested Changes:**

- Given existing literature on machine learning (ML) for combinatorial optimization (CO) and decision optimization (DO), and such, it is not clear why BPP needs a separate survey. What is it about BPP that is not subsumed in the ML for CO surveys and such. This relates to both **W1** & **W2**, and is critical in my opinion.

- Any changes to mitigate weaknesses **W3** and **W4** is also critical in my opinion.

- Equation 2 requires more description for ML audience: if $\text{cost}(\sigma)$ is the minimum possible number of used bins, why is there the condition $\text{cost}(\sigma) = n$, where $n$ is the number of items. This is confusing, and implying that the minimum cost is when each item is put in its own bin, which is probably not the minimum. This is either a mistake or needs a deeper explanation.

- Section 2.2, describing existing BPP solutions, goes into too much technical detail, probably meant for readers familiar with BPP, but not for a wider ML audience. Many of these concepts and explanations could use visual examples for better understanding. Alternately, it would be good to evaluate if there is a need to understand these concepts for ML for BPP. If there is no need, maybe such technical (but not descriptive enough) details can be excluded from the survey.

- Figure 1 (page 2) is brought up at the end of Section 2 (page 6) without much context -- it just says "Comparison is shown in Figure 1". It is not clear what is being compared in Figure 1. It appears to be a single pipeline. Figure 1 is also not well explained in the caption -- the pieces are somewhat vague with acronyms and names which are not defined anywhere near the Figure 1 or Section 2 where it is referred to. This presentation should be significantly improved.

- I think it might be useful to contextualize the different components in the state/action representation (like "height map", "item tensor", etc) and how they relate to the formal BPP problem presented. How are the objectives and constraints handled? This should be better presented

- In the RL setup, how are the different components on the BPP problem mapped to different parts of the RL. What maps to rewards? How are constraints handled? What are state and action spaces and how they relate to the BPP (and the optimization variables in it)? Further details should be provided here for better presentation.

- In 3.2, EMS is mentioned without being introduced. Same goes for various acronyms in the text and Figure 1. Can they be appropriately introduced in the survey?

- The selection of the benchmark and the methods for evaluation needs better motivation in Section 5. The tuning argument seems unclear. Is the tuning done in a way that the method is no longer applicable to other BPP problems. In that case, this is another important aspect of the BPP problem and the use of ML for BPP that should be covered in the survey. Alternately, if the methods are applicable but need to be tuned, hyperparameter optimization is now a very standard procedure in machine learning, and could have been employed here for evaluating other methods. It is critical to clear this issue.

- There are various surveys that compare methods (that were previously not part of a comparitive study) on benchmarks (that the methods were not previously evaluated on). Based on the results of the extended comparitive evaluation, the surveys highlight what aspects of different methods make them more or less performant. A similar treatment here would be a very beneficial addition to this survey highlighting what works for the BPP problem and what does not (for example, when is a supervised learning based scheme better, or what state representations are most beneficial for a reinforcement learning based scheme).

**Strengths And Weaknesses:**

## Strengths:

- This suvery attempts to bring the problem of multi-dimensional Bin Packing to the attention of the machine learning community. Given that almost all of the machine learning based methods listed in this survey are not published in the usual machine learning conferences, this survey is a good bridge to expose machine learning researchers to this problem.
- The survey puts together all the available benchmarks, making it easier for the researchers to thoroughly evaluate their methods.

-------------------

## Weaknesses:

- **W1**: In my opinion, the main weakness is that the goal of the survey is unclear. If we want the readers to know that there is this combinatorial optimization problem -- the Bin Packing problem -- that can be solved with reinforcement learning, this is something that seems to be already covered by existing ML+CO (machine learning for combinatorial optimization) surveys. If this survey is just saying "there are these papers on machine learning for BPP, go read them" it is not a good survey in my opinion, and some of that is already done with the ML+CO surveys.

- **W2**: The main differentiating factor between the broader ML+CO topic and the specific topic of machine learning for Bin Packing problem is the way the problem is represented. For example, in the use of reinforcement learning, the main differentiators are how the state spaces and actions representations are crafted. This survey does not appropriately highlight this distinction.

- **W3**: This survey just presents the Bin Packing problem in a mathematical form in equation (1), and briefly details the different aspects. To better understand the role of machine learning in this combinatorial optimization problem, it would be useful to tease out the salient components of the mathematical form that machine learning can leverage or help with. This survey does not cover this to the best of my understanding.

- **W4**: In general, the use of machine learning for combinatorial optimization requires a discussion on the learning setup. For example, in many cases, there is a distribution of "similar" combinatorial optimization problems of the same form that one needs to repeatedly solve, and the machine learning uses some instances from this distribution to then help solve future instances more efficiently. In some other cases, the solution of a single problem is treated as an episode in a reinforcement learning setup, and the agent is trained over many episodes. This survey does not discuss the learning setup at all. There might be a single setup or multiple setup, but it seems important to have that (or those) spelled out in a survey.

---

### Review · Reviewer_W1Nh · 2024-01-24

**Summary Of Contributions:**

This paper is a survey on 3D bin packing using ML methods. The paper does a systematic review of both classical and modern ML-based methods for 3D bin packing. The paper also introduces 4 types of benchmark datasets, and conducts experiments to compare recent ML-based 3D bin packing algorithms.

**Audience:**

Yes

**Broader Impact Concerns:**

None.

**Claims And Evidence:**

Yes

**Requested Changes:**

- The second paragraph, “we can never apply the exact method” — maybe “never” is too strong? Couldn’t there be “easier” problem instances that may be solved exactly? For instance, there are integer programming solvers (which may be used for BPP), and good solvers can solve moderately large instances.

- “Eliiyi & ELIIYI” why the second one is in capital letters

- I think that the definition of the problem is not explained well enough. The authors try to define the problem using the big equation (1) in Section 2. However, I think that it would be more clear if the author could first list the constraints in words and/or simple notations, and then relate that to the formulation in (1). This is because (1) contains nontrivial equations, such as (1f), which is not easy to understand without knowing what it try to formulate (and it is appreciated if the author could explain this in detail). As an analogue, think about TSP, people usually define TSP in words and/or simple notations, but they rarely just say the TSP is defined using some complicated integer linear program.

- A concrete question to (1): what’s \Gamma? You said this is number of usable bins, but you also said the goal is to minimize the number of bins used (which you use \gamma) — this is contradicting. Also, it seems \Gamma is not quantified as a variable in (1), so does this mean \Gamma is part of the input? This is very confusing.

- In the first paragraph of Section 2.2, you ref to Figure 1 for online vs offline settings. However, Figure 1 is actually about a typical pipeline of ML method.

- At the bottom of page 4, the notion of “performance ratio” is introduced. However, this is inside the subsection “Problem Variants”, but it seems this notion is already important/fundamental to the base problem. It is thus suggested to move this to the beginning of the entire section (or some other place that makes more sense).

- The paragraph of “Placing Candidates” in page 5 — I have difficulty understanding the logic. You mentioned Martello et al. (2000) defined the notion of “corner point”, Crainic et al. (2008) defined “extreme point”, and Goncalves & Resende 2013 defined EMS. But what’s all these three notions about? How are they related to the “placing candidate”, especially why these notions are useful for the algorithm (e.g., used in which step)? I don’t find this clearly explained, and that’s why I’m confused.

- I have a general comment about Section 2.2. This section is called “Problem Variants”, but the contents are not only about the “problems”, and they are also about “how to solve” the problems. It’s suggested to better separate the discussion to make the logic clear.

- You mentioned a benchmark called “PackIt”. This actually has a virtual environment. Could you please elaborate on how this virtual environment helps in benchmarking the algorithms?

- In the experiments, why DPPDRL does not generate a result for the “without stability” case?

- In the experiments, how to interpret the running time? It looks very small to me, so I consider this very efficient. However, is it possible to use more time to generate better solutions?

- I understand that the experiments only focus on online algorithms. However, there seems to be a lot of such algorithms available. It might make sense to add more as baselines.

- It would be great to add the MIP (or some other accurate offline algorithm) as a baseline in the experiment since it gives the offline optimal solution, so that it is easier to justify the performance of online algorithms. Moreover, it also offers a baseline for the running time.

**Strengths And Weaknesses:**

Strength:

- The topics covered seem to be comprehensive
- The challenges and progress of the area have been discussed
- The benchmark and experiments can be very useful for future study

Weakness:

- The experiments are weak and not very convincing
- In general, I find the discussion of previous works is technical-heavy, and high-level ideas/results are less discussed

---

### Review · Reviewer_YKzN · 2024-02-27

**Summary Of Contributions:**

This paper presents a review of (a few) recent ML-based approaches to the (online) 3D bin packing problem with an attempt to classify the different methods.

The authors describe four datasets that might be used for benchmarking new approaches to this problem.

A single table (with literally four lines) is reported as “computational results” for comparing four methods.

**Audience:**

Yes

**Broader Impact Concerns:**

The are not ethical implications for this paper.

**Claims And Evidence:**

No

**Requested Changes:**

The survey should be completely rewritten and it should focus on online bin packing problems, and more specifically on online 3D bin packing problems, where the ML-based approaches are indeed promising.

The authors should make an effort to highlight the key features of each paper from the literature in a more systematic and principled way. For instance, is a given work relevant because it proposes a better encoding of the search 3D space? Is a paper interesting because it permits to infer hidden patterns in the distribution of the items that are revealed only in the “online phase”? Does the paper improve a specific (heuristic) step of a known algorithm by “learning” the best strategy for selecting and packing the next item?

While these types of questions are partially answered in the current version of the paper, there is not a true systematic review that helps the reader to get a clear idea of the state-of-the-art on ML-based approaches for online packing problems (without having to read the original papers). In other words, the survey should be also a sort of tutorial that could motivate the readers to tackle the computational challenges that arise in these kinds of online packing problems.

Tables 1 and 2 could be a good starting point, but they are too dense, and too many details remain unexplained.

The computational results should be more detailed and informative. The concept of “stability” is completely unclear.

Finally, the list of references should be more precise (e.g., report the page numbers of every journal paper).

**Strengths And Weaknesses:**

The main strength of this paper is the topic since the 3D Bin Packing problem is challenging and it has several industrial applications. Specifically, the “online” version is relevant for ML-based approaches where, at least in principle, it could be possible to “learn” some hidden distributions (or patterns) of the online version of the problem.

However, the paper has unfortunately several weaknesses. First, the writing is below the standard of TMLR, since it contains too many sloppy sentences (starting from the abstract: “[ML], … which is famous for its efficiency” (which type of efficiency?), or page 2 (“the imperative need for efficient … algorithms”, we all need efficient algorithms). There are several sentences of this type, and they do not add any value to a technical (scientific) paper (as another example, the three small paragraphs on page 2, entitled “Efficiency”, “Data-driven characteristic” and “less reliance on domain knowledge” are redundant and useless).

The paper should be better organized and should focus on a single problem, that is the “online 3D Bin packing problem” (with or without additional side constraints), which is the main version of bin packing where machine learning can be exploited to design better solution algorithms. Only for that problem, the authors should give more details on the relevant features (the main contribution of each cited paper), and run more significant computational experiments to compare the different approaches. The results should highlight the strengths and weaknesses of each method, compared to the others.

In its current form, this paper looks more like a long list of references related to different aspects of the bin packing problem, and it is not a “systematic review”. Just two examples for improving the paper:

1. You cite the work of Christensen et all 2017, which contains an exemplary systematic review of approximation and online algorithms for the multidimensional bin packing problem. That survey closes with 10 open problems: how can the use of ML help in solving any of those problems? How can an ML algorithm help in overcoming (if possible) the negative approximation results discussed in that paper?

2. The first paper you report in Table 1 was rejected at ICRL 2020: https://openreview.net/forum?id=BkgTwRNtPB
Why do you include a paper that was rejected? While this is possible that the paper is interesting despite the rejection, it should be motivated in the paper why that paper deserves to be cited.


The model introduced in Section 2 should be clarified. For instance, there is no description/comment for constraint (1-f). Moreover, it is unclear why you consider 6 rotations instead of only 3, since on each axis you can shift after the rotation unless you have an additional constraint on the direction of a single item (besides the 3 dimensions length, width, and height).

What is the purpose of Figure 2? Just to show 4 items in a single bin? Maybe a more elaborate illustration on the challenge of packing in 3D would be better. Anyway, for the two cubes, I will not talk about “rotation”: the three dimensions are equal and their "rotation" does not have an impact on the volume occupied.

What is the purpose of the dataset described in Section 4 if then the methods are compared in Section 5 using only the CSD dataset? What should the reader infer from the (micro)table of Section 5? Why report in Table 3 the details of the random dataset if later is unused in Section 5?

Moreover, in Section 5, there is no explanation of the type of experiments, the software and libraries used, the computer used, the training and validation runtime, and accuracy.

There is a rather vague sentence “We do not compare ML-related methods with commercial solvers, because….” First of all: which commercial solver? For 3D bin packing?
What do you mean in the sentence “Therefore, it cannot be solved by MIP”? MIP solvers are a key component of several state-of-the-art algorithms (e.g., to run math-based heuristics as in Elhedhli, 2019).


Finally, we suggest the authors look at the results of the Roadef Computational challenge, 2022 edition, supported by Renault, which was related to an offline 3D Bin packing problem arising as an application of truck loading in logistics:
https://github.com/renault-iaa/challenge-roadef-2022/tree/main

---

### Note · Authors · 2024-02-28

**Comment:**

Thanks for your comments. We will revise our paper according to the reviews before submitting it again.

**Withdrawal Confirmation:**

I have read and agree with the venue's withdrawal policy on behalf of myself and my co-authors.